©c Author(s) 2022. CC BY 4.0 License.



# Tracing the Snowball bifurcation of Aquaplanets through time reveals a fundamental shift in critical-state dynamics

Georg Feulner[1], Mona Sofie Bukenberger[1,2], and Stefan Petri[1]

[1]Earth System Analysis, Potsdam Institute for Climate Impact Research, Member of the Leibniz Association, Potsdam, Germany
[2]Institute for Atmospheric and Climate Science, ETH Zürich, Switzerland

**Correspondence:** Georg Feulner (feulner@pik-potsdam.de)

**Abstract.** The instability with respect to global glaciation is a fundamental property of the climate system caused by the positive ice-albedo feedback. The atmospheric concentration of carbon dioxide ($CO_2$) at which this Snowball bifurcation occurs changes through Earth's history, most notably because of the slowly increasing solar luminosity. Quantifying this critical $CO_2$ concentration is not only interesting from a climate dynamics perspective, but also an important prerequisite for understanding past Snowball Earth episodes as well as the conditions for habitability on Earth and other planets. Earlier studies are limited to investigations with very simple climate models for Earth's entire history, or studies of individual time slices carried out with a variety of more complex models and for different boundary conditions, making comparisons and the identification of secular changes difficult. Here we use a coupled climate model of intermediate complexity to trace the Snowball bifurcation of an Aquaplanet through Earth's history in one consistent model framework. We find that the critical $CO_2$ concentration decreases more or less logarithmically with increasing solar luminosity until about 1 billion years ago, but drops faster in more recent times. Furthermore, there is a fundamental shift in the dynamics of the critical state about 1.2 billion years ago, driven by the interplay of wind-driven sea-ice dynamics and the surface energy balance: For critical states at low solar luminosities, the ice line lies in the Ferrel cell, stabilised by the poleward winds despite moderate meridional temperature gradients under strong greenhouse warming. For critical states at high solar luminosities on the other hand, the ice line rests at the Hadley-cell boundary, stabilised against the equatorward winds by steep meridional temperature gradients resulting from the increased solar energy input at lower latitudes.

## 1 Introduction

Today, the climate of planet Earth is in a state which is neither too hot nor too cold for life, with the ice line being far from the equator. The position of the ice line is ultimately determined by the planetary energy balance, in particular the incoming solar radiation and the concentration of greenhouse gases in the atmosphere, as well as the ice-albedo feedback and meridional heat transport in Earth's climate system (e.g. North et al., 1981).

One of the consequences of the positive ice-albedo feedback is the phenomenon that over a range of boundary conditions more than one state of Earth's climate can be stable for the same levels of incoming solar radiation and greenhouse gases. So even with the current solar luminosity and atmospheric composition, Earth might as well rest in a state a lot less welcoming to





living beings – a fully glaciated Snowball state with surface temperatures multiple tens of degrees lower than today (e.g. North, 1990). There would be no liquid water at Earth's surface, a necessary condition at least for complex life as we know it.

    For even lower solar luminosities and/or greenhouse-gas concentrations, a bifurcation point in phase space is reached. This means that for a certain set of parameters, the system ends up with only the Snowball state being stable. If the current climate cooled down, the ice line would descend slowly towards the equator at first. But at some point, the system would tip: Earth

would rapidly cool down, ending up in a global glaciation (Öpik, 1953). The Snowball instability crossed in this case is fundamental to the climate system and ultimately caused by the positive ice-albedo feedback.

    The first to quantify the Snowball instability in terms of the critical solar luminosities for given system parameters were Budyko (1969) and Sellers (1969) who analysed the Snowball bifurcation with one-dimensional energy balance models (EBMs, North et al. 1981). These EBMs are built upon the principle of energy conservation at a given latitude and consist of either

a time-dependent differential equation for the temperature as a function of latitude or a time-independent equation assuming that the system is in equilibrium. Budyko and Sellers found that even a small decrease in solar luminosity would suffice for an Earth with today's system parameters (in terms of the greenhouse effect) to end up in a state of complete glaciation.

    Despite their simplicity, many authors have used EBMs – often incorporating various modifications and extensions – ever since the pioneering studies by Budyko (1969) and Sellers (1969), especially in order to understand how various factors

influence the Snowball instability and the climate system's phase space properties (Faegre, 1972; Schneider and Gal-Chen, 1973; Held and Suarez, 1974; North, 1975a, b; Gal-Chen and Schneider, 1976; Ghil, 1976; Drazin and Griffel, 1977; Lindzen and Farrell, 1977; Cahalan and North, 1979; North and Coakley, 1979; North et al., 1981, 1983; North, 1990; Huang and Bowman, 1992; Ikeda and Tajika, 1999; Shen and North, 1999; Rose and Marshall, 2009; Roe and Baker, 2010).

    In terms of Earth's long-term habitability, the Snowball bifurcation is particularly relevant in light of the fact that the solar

luminosity was considerably lower in the past (e.g. Gough, 1981). The time evolution of the critical $CO_2$ concentration required to prevent global glaciation has typically been studied with radiative-convective models (RCMs; Ramanathan and Coakley, 1978) rather than EBMs. However, in RCMs, the ice-albedo feedback is not taken explicitly into account, but has typically been considered by requiring a global mean surface temperature well above the freezing point of water. Examples for such investigations can be found in Kasting (1987) or von Paris et al. (2008), for example.

In addition to considerations of planetary habitability, open problems related to specific time periods in Earth's history are the second important reason why one should be interested in the Snowball instability. The most relevant geologic eons in this respect are the Archean (about 4 to 2.5 Ga, 1 Ga = $10^9$ years ago) with evidence for habitable conditions on Earth despite a considerably fainter young Sun (Feulner, 2012; Charnay et al., 2020) and the Proterozoic (about 2.5 to 0.54 Ga) for which there is geological evidence for near-global glaciations both in the beginning (Tang and Chen, 2013) and towards the end of

the eon (Hoffman and Schrag, 2002).

    During the Archean, the luminosity of the young Sun was reduced by 20–25% as compared to today. Results from EBMs and RCMs suggest that such a large reduction in the incoming solar radiation should have turned Earth into a Snowball, yet the geologic record suggests the presence of liquid water. This puzzling discrepancy, known as the *Faint Young Sun Paradox* (Feulner, 2012; Charnay et al., 2020), has led to considerable interest in quantifying the Snowball bifurcation point on early





Earth. For four decades, climate-modelling work on the Faint Young Sun Paradox has been dominated by radiative-convective climate models essentially neglecting the ice-albedo feedback. More recently, however, a number of research groups have published first results from three-dimensional climate models investigating solutions for the Faint Young Sun paradox (Kienert et al., 2012, 2013; Charnay et al., 2013; Wolf and Toon, 2013; Kunze et al., 2014; Le Hir et al., 2014). However, all of these model studies are limited either by simplifications in their atmosphere or in their ocean/sea-ice components and use a variety of different assumptions and boundary conditions, leading to a large spread in simulated critical $CO_2$ concentrations.

In contrast to the Archean, there is geologic evidence for low-latitude glaciations during the Proterozoic (Tang and Chen, 2013; Hoffman and Schrag, 2002), so the central question becomes *What are the conditions required for Snowball events?* rather than *What prevented Earth from freezing over completely?* This problem has been investigated in a number of modelling studies, in particular for the Neoproterozoic (e.g. Chandler and Sohl, 2000; Poulsen et al., 2002; Donnadieu et al., 2004; Pierrehumbert et al., 2011; Voigt et al., 2011; Liu et al., 2013; Feulner and Kienert, 2014; Feulner et al., 2015; Liu et al., 2017), yielding critical $CO_2$ concentrations from below 40 ppm to about 700 ppm depending on model type and boundary conditions.

Despite the relevance of the phenomenon for our understanding of past climate states and planetary habitability, there has – to our knowledge – not yet been an attempt to study the Snowball instability throughout Earth's history within one consistent framework and using a more complex climate model. At one end of the spectrum, earlier studies comprise conceptual investigations with very simple climate models like RCMs or EBMs. The latter in particular help to understand the principles of the instability and changes in phase space. At the other end of the spectrum, there are many investigations of single events in Earth's history with climate models of various complexities, ranging from models of intermediate complexity to atmosphere-ocean general circulation models (AOGCMs), and with a variety of different boundary conditions. These studies provide detailed insights about the time slices investigated, but the lack of uniform simulation design, model architecture and boundary conditions makes it hard to compare them to each other or to study the evolution of the Snowball instability with time.

In this study, we want to bridge the gap between studies of the Snowball instability for single time slices with complex models and conceptual investigations of its time evolution. To this end, we use an Earth-system model of intermediate complexity (EMIC, Claussen et al. 2002) in an Aquaplanet configuration to scan for the Snowball bifurcation point for time slices spanning the last 4 billion years, thus quantifying the time evolution of the bifurcation and identifying a qualitative shift in critical state dynamics.

This article is organised as follows. In Sect. 2, we describe the coupled climate model used to scan the Snowball bifurcation of Aquaplanets through Earth history, the boundary conditions as well as design of our numerical experiments. In Sect. 3, we present the Snowball bifurcation for Aquaplanets through time and compare our results to earlier studies (Sect. 3.1). Furthermore, we describe global properties of the critical states for the different time slices (Sect. 3.2) and discuss the two different dynamical regimes for the critical state (Sect. 3.3). Finally, in Sect. 4 we summarise the major findings of our work in the context of earlier studies, discuss limitations and outline potential future research.





## 2   Methods: Coupled climate model simulations

### 2.1   Model description

Scanning for the Snowball bifurcation for more than a dozen time slices throughout Earth's history requires a relatively fast
coupled climate model. We employ the Earth-system model of intermediate complexity CLIMBER-3$\alpha$ (Montoya et al., 2005).
CLIMBER-3$\alpha$ consists of a modified version of the ocean general circulation model (OGCM) MOM3 (Pacanowski and
Griffies, 1999; Hofmann and Morales Maqueda, 2006) with a horizontal resolution of $3.75° \times 3.75°$ and 24 vertical levels,
a dynamic/thermodynamic sea-ice model (Fichefet and Morales Maqueda, 1997) at the same horizontal resolution and a fast
statistical-dynamical atmosphere model (Petoukhov et al., 2000) with a coarse horizontal resolution of $22.5°$ in longitude and
$7.5°$ in latitude.

We emphasize that the sea-ice model explicitly takes into account sea-ice dynamics, a factor which has been found to be of
crucial importance for the Snowball bifurcation (Lewis et al., 2003, 2007; Voigt and Abbot, 2012). The Snowball bifurcation
also critically depends on cryosphere albedo (e.g. Yang et al., 2012a). Our model uses clear-sky albedo values (averaged over all
wavelengths) of 0.50 and 0.40 for freezing and melting sea ice, and 0.75 and 0.65 for cold and warm snow, respectively. Albedo
values for ultraviolet+visible light are 0.30 larger than near-infrared albedos, and a partitioning of 60% (ultraviolet+visible)
and 40% (near-infrared) is assumed. The effects of snow cover on sea ice are explicitly taken into account.

The main limitations of the model relate to its simplified atmosphere component. Particularly relevant for this study are the
coarse spatial resolution, the highly parameterised vertical structure and the simplified description of large-scale circulation
patterns, including the fixed annual-mean width of the Hadley and Ferrel cells. We note, however, that the Snowball bifurcation
points derived for Neoproterozoic time slices with our model (Feulner and Kienert, 2014) agree very well with the those from
a state-of-the-art atmosphere–ocean general circulation model (AOGCM) with similar albedo values (Liu et al., 2013). The
impact of model limitations on our results will be discussed in Sect. 4.

### 2.2   Boundary conditions and design of numerical experiments

To facilitate comparison between the different time slices we chose an Aquaplanet configuration without any continents. The
ocean topography was generated by randomly assigning an ocean depth to each grid cell using a Gaussian probability distri-
bution with a mean depth of 3000 m and a variance of 450 m. For each grid cell, the resulting random depth value was then
assigned to the corresponding vertical level of the ocean model's grid. We chose to have an ocean with varying depth rather
than a flat ocean floor in order to avoid potential numerical instabilities.

The Snowball bifurcation point is derived for a total of 18 time slices ranging from today to 3600 Ma (million years ago),
see Table 1. The solar constant was scaled based on the approximation formula from Gough (1981), assuming a present-day
value of $S_0 = 1361$ W/m$^2$ (Kopp and Lean, 2011) and an age of the Sun of 4570 Ma (Bonanno et al., 2002). Orbital parameters
were set to a circular orbit with obliquity $23.5°$. For each time slice, a number of equilibrium simulations were run for different
$CO_2$ concentrations bracketing the Snowball bifurcation (see Table 1). In addition, we have run model experiments at two
fixed levels of $CO_2$ (10,000 ppm and 10 ppm) and decreasing solar luminosities of 1140 W/m$^2$, 1130 W/m$^2$, 1125 W/m$^2$ and





**Table 1.** Overview of simulation experiments with the age of each time slice, the value of the solar constant $S$ used in the simulations and the atmospheric $CO_2$ concentration of not fully glaciated and Snowball states.

| Age (Ma) | $S$ (W/m$^2$) | Non-Snowball states $p$CO$_2$ (ppm) | Snowball state $p$CO$_2$ (ppm) |
|---|---|---|---|
| 0 | 1361 | 277, 1, 0.9, 0.8, 0.7, 0.5, 0.3, 0.1 | 0 |
| 150 | 1343 | 4, 3 | 2 |
| 300 | 1327 | 30, 15, 12, 11 | 10 |
| 500 | 1304 | 60, 50, 45 | 40 |
| 700 | 1285 | 110, 100 | 90 |
| 900 | 1261 | 600, 500, 400, 350, 300, 290, 280, 270, 260, 250, 240 | 230 |
| 1050 | 1246 | 400, 390, 380 | 370 |
| 1200 | 1231 | 600, 590, 580, 570 | 560 |
| 1350 | 1217 | 900, 850, 830, 828, 826, 824, 822 | 820 |
| 1500 | 1203 | 1400, 1300, 1250, 1200, 1180, 1170, 1165, 1163, 1161 | 1160 |
| 1650 | 1190 | 1900, 1850, 1800, 1700, 1650, 1640, 1630 | 1620 |
| 1800 | 1176 | 3000, 2900, 2800, 2700, 2650, 2620 | 2610 |
| 2100 | 1149 | 5000, 4980, 4960, 4950, 4940 | 4920 |
| 2400 | 1125 | 12500, 11500, 11000, 10000, 9900 | 9800 |
| 2700 | 1100 | 20000, 19500, 19300, 19200 | 19100 |
| 3000 | 1078 | 35000, 34000, 33000, 32700, 32600 | 32500 |
| 3300 | 1055 | 60000, 56500, 56000, 55500, 54000, 53000 | 52000 |
| 3600 | 1034 | 80000, 79500, 79000, 78600, 78200, 78000, 77700 | 77300 |

1120 W/m$^2$ for 10,000 ppm, and 1334 W/m$^2$, 1329 W/m$^2$ and 1324 W/m$^2$ for 10 ppm respectively. For the lowest value of the solar constant in each of these cases the model entered a Snowball state.

The total atmospheric pressure was kept constant at 1 bar in all simulations. Most simulations were initialised from a warm, ice-free state with idealised symmetric present-day ocean temperature and salinity fields. In many cases simulations pinpointing the Snowball bifurcation point were branched from runs with higher $CO_2$ concentrations. All simulations were integrated for

at least 2,000 model years after the last change in $CO_2$ concentration to allow the ocean approaching equilibrium conditions.





## 3 Results

### 3.1 The Snowball bifurcation for Aquaplanets through time

The critical $CO_2$ concentration for the Snowball bifurcation through Earth's history as derived from the Aquaplanet simulations with CLIMBER-$3\alpha$ is shown in Figure 1. As expected, there is excellent agreement between the bifurcation points derived at

the two fixed $CO_2$ levels and the closest corresponding simulation derived at fixed values of the solar constant. In other words, there is no fundamental difference between scanning for the critical values in the horizontal and the vertical direction in the diagram. In the figure, the values are also compared to proxy estimates for the past $CO_2$ concentration in Earth's atmosphere and to earlier modelling studies for individual time slices, the latter differentiated by model type.

With solar luminosity increasing over time, the critical $CO_2$ concentration in our Aquaplanet simulations falls from $\sim$

$10^5$ ppm to below 1 ppm between 4 Ga and today. This decrease is more or less logarithmic from 4 Ga to 1 Ga, with more steeply falling $CO_2$ levels in more recent times. The logarithmic decrease with linearly increasing insolation can be understood in terms of the well-known logarithmic behaviour of the $CO_2$ radiative forcing (Huang and Bani Shahabadi, 2014). The downturn of the critical $CO_2$ concentration for solar luminosities approaching the modern value can be attributed to three factors: (1) The steadily increasing incoming solar radiation (which shows a strong variation with latitude) leads to a more

positive surface radiation balance in particular over the open ocean areas despite a relatively weak greenhouse warming (which is more uniformly distributed), making sea-ice formation at low latitudes increasingly difficult. (2) Even at the very low global mean surface air temperatures of $\sim -15°C$ of the critical states (see Figure 2), there is a baseline greenhouse warming due to water vapour which becomes significant at very low $CO_2$ levels. This is also facilitated by the fact that the maximum of the thermal emission spectrum is shifted towards longer wavelengths and thus into the $H_2O$ rotational bands because of the lower

temperature. (3) Finally, there is also a warming contribution from cloud radiative effects.

The critical $CO_2$ concentrations as a function of the solar constant $S$ derived from our Aquaplanet simulations can be approximated by the following formula ($S_0$ is the present-day solar constant):

$$pCO_{2,\,crit} = a_1 \exp\left[a_2 \left(a_3 - \frac{S}{S_0}\right)^{a_4}\right] \tag{1}$$

Fitting this function to the points derived from the Aquaplanet simulations yields the following parameters: $a_1 = (0.0836 \pm$

$0.0721)$ ppm, $a_2 = 26.6 \pm 0.7$, $a_3 = 1.0 \pm 0.00002$ and $a_4 = 0.475 \pm 0.051$, providing a good approximation over the entire range of solar luminosities, see Figure 1.

Before discussing our results in the context of previous model studies for several key time periods in Earth's history in Sect. 3.1.2, we will describe and discuss more general features which can be derived from this synthesis and earlier studies.

### 3.1.1 General observations on the Snowball bifurcation

**The presence of continents and ice sheets makes global glaciations easier.** For models of similar design, the presence of continents makes Earth more susceptible to glaciation as compared to an Aquaplanet configuration, predominantly due to the





higher surface albedo of land areas compared to open oceans and the lower water vapour content of the atmosphere (Poulsen et al., 2002; Liu et al., 2013; Kunze et al., 2014). The Aquaplanet simulation for a modern solar constant and 277 ppm of $CO_2$, for example, has a global and annual mean surface air temperature of $19.4°C$ compared to $15.1°C$ in a pre-industrial

simulation using our model (Feulner, 2011). Similarly, our simulations with Neoproterozoic continents (Feulner and Kienert, 2014) indicate slightly higher critical $CO_2$ values than the Aquaplanet simulation at similar solar luminosity, although the difference is small in this case due to the low albedo of bare land assumed in Feulner and Kienert (2014). Note that in the extreme case of a fully land-covered planet global glaciation becomes more difficult because the drier atmosphere leads to reduced cloud and snow cover and thus a lower albedo compared to the Aquaplanet case (Abe et al., 2011).

It has also been shown already that the presence of tropical ice sheets shifts the Snowball bifurcation point to values $\sim 3$–$10$ times higher (Liu et al., 2017) than without ice sheets (Liu et al., 2013). The combined effect of continents and polar ice sheets is also the most likely cause for the lower critical $CO_2$ concentrations of the Aquaplanet simulations for modern boundary conditions (Yang et al., 2012a, b, c) and the Late Paleozoic Ice Age (Feulner, 2017). In addition, the simulations in Feulner (2017) were carried out for a glacial orbital configuration rather than the circular orbit used in the Aquaplanet simulations.

**Models with simplified oceans and without sea-ice dynamics are more stable.** Studies carried out with atmospheric general circulation models (AGCMs) coupled to mixed-layer ocean models and without dynamic sea ice tend to predict lower values for the Snowball bifurcation point (see Figure 1). This could be either due to the prescribed ocean heat transport in these models or the lack of sea-ice dynamics. Indeed, the fact that models without sea-ice dynamics are artificially stable with respect to the Snowball bifurcation has been noted before (Lewis et al., 2003, 2007; Voigt and Abbot, 2012).

**The glaciation threshold depends on sea-ice and snow albedo.** Even for models of the same design there is considerable spread in the values for the Snowball bifurcation point for similar boundary conditions (see Figure 1). Differences in cloud radiative forcing and simulated heat transport in the atmosphere and the oceans can contribute to this spread, however, the predominant causes are differences in sea-ice and snow albedo values and parametrisations (Yang et al., 2012c). These have been identified as the cause of the difference between the bifurcation points derived with CCSM3 (Yang et al., 2012a) and

CCSM4 (Yang et al., 2012c) for the same present-day boundary conditions, for example.

### 3.1.2 Comparison with earlier modelling studies for selected time periods

**Modern Snowballs.** The Snowball bifurcation point for modern boundary conditions has been quantified with an AGCM in terms of reduced $CO_2$ by Romanova et al. (2006) and with AOGCMs in terms of a reduced solar constant by Voigt and Marotzke (2010) and by Yang et al. (2012a, b, c). For pre-industrial greenhouse gas concentrations, the AOGCM studies put

the bifurcation point in the ranges 91–94% (Voigt and Marotzke, 2010), 89.5–90% (Yang et al., 2012a) and 91–92% (Yang et al., 2012c) of the present-day solar constant, comparing well to about 91% of today's solar constant in our simulations (see Figure 1). For a reduction in $CO_2$ and a fixed present-day solar constant, the bifurcation point in our Aquaplanet simulations is below a $CO_2$ concentration of 0.1 ppm. This agrees well with the AGCM study by Romanova et al. (2006) where their model is not in a Snowball state at their lowest $CO_2$ concentration of 1 ppm.



**Proterozoic.** For the Neoproterozoic, earlier model studies indicate critical $CO_2$ concentrations from below 40 ppm in an AGCM simulation (Chandler and Sohl, 2000) to about 100–700 ppm in AOGCMs and OGCMs with sea-ice dynamics (Poulsen et al., 2002; Voigt et al., 2011; Liu et al., 2013; Feulner and Kienert, 2014; Liu et al., 2017). (The reason for the higher value found by Lewis et al. (2003) remains unclear.) In their studies on modern Snowballs, Yang et al. (2012a, c) find critical $CO_2$ concentrations of about 20 ppm to 100 ppm depending on model version for a solar constant representative of the Neoproterozoic, i.e. reduced by 6% compared to its modern value. In our Aquaplanet configuration, the corresponding value at 700 Ma is $95 \pm 5$ ppm and thus well within the typical range of the models with ocean and sea-ice dynamics. With the exception of Chandler and Sohl (2000) and Yang et al. (2012a), our value is somewhat lower than the one derived in other studies, as one would expect for a configuration without continents or ice sheets and for the snow and sea-ice albedo values employed in our model (see also Sect. 2.1 and Sect. 3.1.1). Furthermore, there is also excellent agreement with the bifurcation point quantified using an AOGCM for an earlier time slice at 1 Ga by Fiorella and Sheldon (2017), where our values for 900 Ma and 1,200 Ma are again compatible with the lower range of their estimate, see Figure 1.

**Archean.** In many ways, the situation is most complicated for the Archean where the Snowball bifurcation has been quantified in a number of model studies in the context of the Faint Young Sun Paradox (Feulner, 2012; Charnay et al., 2020). So far, there are no AOGCM studies for the Archean yet. The bifurcation points from Kienert et al. (2012), quantified with an earlier version of the model used in this work, are higher than the ones for the Aquaplanet configuration, mainly caused by the higher cryosphere albedo values used in Kienert et al. (2012) and the different boundary conditions, in particular the higher rotation rate. All other model studies use AGCMs with simplified ocean components and sea-ice models without dynamics (Charnay et al., 2013; Wolf and Toon, 2013; Kunze et al., 2014; Le Hir et al., 2014; Teitler et al., 2014). The critical $CO_2$ values from these studies are generally (and in many cases significantly) below the values derived for the Aquaplanets in this paper. While we cannot rule out a contribution from the simplified atmosphere component used in our model, the lack of full ocean and in particular sea-ice dynamics in the other studies is likely to be a significant factor in explaining this discrepancy. In the end, this question can only be fully answered by running AOGCM simulations with Archean boundary conditions which is beyond the scope of the present work.

## 3.2 Global characteristics of Aquaplanets critical states

In the following, we will have a more detailed look at global characteristics of the climate states close to the Snowball bifurcation, i.e. the stationary states for all investigated solar luminosities with the lowest concentration of $CO_2$ in the atmosphere for which the system does not fall into a Snowball state.

Let us first take a look at the global annual mean surface air temperature and the mean sea-ice fraction of the critical states for the different time slices shown in Figure 2, where the averages are taken over the last 100 years of each simulation. Whereas the critical $CO_2$ concentrations as a function of solar luminosity would indicate a fairly smooth change over time (see Figure 1), there is a marked shift in both the global annual mean surface air temperature and the mean sea-ice fraction: Critical states at low solar luminosities consistently have global mean temperatures of about $-2°C$ and average sea-ice fractions of about 33%, whereas critical states at higher solar luminosities exhibit much lower temperatures of about $-15°C$ and higher sea-ice



fractions of 50%. In addition, there is a slight cooling trend with increasing solar luminosities for the critical states both at
lower and at higher solar luminosities. Finally, the sequence of critical states with higher sea-ice fractions shows remarkably
little scatter in this quantity, see Figure 2 (b).

The values of 33% and 50% for the global annual mean sea-ice fraction found for the critical states can be translated into
latitudes of the sea-ice margin of 42° and 30°, respectively, if one assumes full ice cover over two symmetric polar caps which
is a reasonable assumption for Aquaplanets. Thus, for lower solar luminosities, the sea-ice margin rests within the Ferrel cell
of the atmosphere's large-scale circulation, whereas for higher solar luminosities it corresponds to the Hadley-cell boundary
(which is fixed at 30° in our simplified atmosphere model). We will therefore refer to the critical states at lower solar luminosity
as *Ferrel states* and to the ones at higher solar luminosity as *Hadley states* for simplicity.

In Figure 2, we focus on the critical states only, i.e. the coldest not fully glaciated state at each solar luminosity. While we did
not find any Hadley states at low solar luminosities, the question remains whether there could be Ferrel states at higher solar
luminosities. This can be investigated by looking at equilibrium states for a range of $CO_2$ concentrations for one particular
time slice at higher solar luminosity. The result for the 900-Ma time slice is shown in Figure 3, where one can see that even for
higher solar luminosities Ferrel states can indeed be stable states at higher atmospheric $CO_2$ levels, as already indicated by the
grey shading in Figure 2.

On the other hand, we can also look for transient Hadley states in simulations for Ferrel-state time slices where the system
falls into the Snowball state. In these simulations, the unstable Hadley state can clearly be identified as a time period with
close to 50% global and annual mean sea-ice cover $f_{\mathrm{sea\,ice}}$. In agreement with the findings of Figure 2, these states (defined by
$0.497 < f_{\mathrm{sea\,ice}} < 0.503$) become more stable with increasing solar luminosity: While the system spends only 70 years in the
transient Hadley state at 3600 Ma, for example, this length more than quintuples to 396 years at 1350 Ma (Figure 4).

In summary, we find a marked shift in the characteristics of the critical states in the time period from 1350 to 1200 Ma:
While a Ferrel state with the sea-ice margin in the middle of the Ferrel cell is stable for all luminosities, a Hadley state with the
ice-line at the Hadley-cell boundary can only be stable for solar luminosities above about 90% of the modern solar constant.
The fact that the critical state at 1800 Ma is a Hadley state already whereas the critical states at 1650, 1500 and 1350 Ma are
Ferrel states again could be due to the sensitivity of the system with respect to small changes, in particular in the transition
period.

## 255 3.3 Dynamics of the Snowball bifurcation of Aquaplanets

In order to understand the different regimes of the critical states we will have a more detailed look at typical Ferrel and
Hadley states. To this end, we select the critical states at 3000 Ma (a Ferrel state) and 900 Ma (a Hadley state) and investigate
the spatial patterns of surface air temperature, sea-ice fraction, and surface winds (Figure 5 and 6). Maps of annual mean
surface air temperatures show that the Ferrel state is considerably warmer at all latitudes (Figure 5). The annual mean sea-ice
distributions exhibit marked differences as well: The Hadley state at 900 Ma has a very well defined, sharp sea-ice margin,
whereas the Ferrel state at 3000 Ma is characterised by a much more fuzzy transition between open ocean and full ice cover





(Figure 6). In both critical states the wind fields show the usual pattern of the large-scale atmospheric circulation, with a high degree of symmetry about the equator as one would expect for Aquaplanet boundary conditions.

These findings are even more evident from the zonal averages of annual mean surface air temperature, sea-ice fraction, and the meridional component of the surface wind for the two critical states shown in Figure 7. In particular we would like to highlight (1) the steeper temperature gradient across the sea-ice margin in the Hadley state, (2) the position of the sea-ice margin in the Hadley state close to the point where the meridional surface winds change their direction, and (3) the latitude of the sea-ice margin in the Ferrel state close to the poleward maximum of the meridional surface wind field.

Indeed, the specific locations of the sea-ice boundaries close to the centre of the Ferrel and at the edge of the Hadley cell immediately suggest that they are influenced by large-scale atmospheric circulation patterns as illustrated in Figure 8: The Ferrel states are stabilised by poleward winds pushing the sea-ice margin away from the equator and transporting relatively warm air towards the ice edge. The Hadley states, on the other hand, are the coldest possible not fully glaciated states because any further cooling would bring sea ice into the Hadley cell where it would be pushed towards the equator by the trade winds, which would also transport cold air towards the sea-ice margin. Since in case of the Hadley states the wind fields are a destabilising factor, these states have to be protected from global glaciation by relatively high surface temperatures within the Hadley cells causing any sea ice which enters the Hadley cells to melt rapidly.

In the following, we will investigate this hypothesis in more detail, in particular with respect to the important question why the Hadley states can be stable only at higher solar luminosities. Answering this question requires, among other things, a closer look at the evolution of the energy balance of the Hadley states. In particular, we would like to understand how and why the surface temperature distribution of the Hadley states changes with increasing solar luminosity. To this end, we follow Heinemann et al. (2009), Voigt et al. (2011), and Liu et al. (2013) in applying a simple equation for the annual mean equilibrium energy balance at each latitude $\varphi$

$$Q(\varphi)\Big(1 - \alpha(\varphi)\Big) = Q_{\mathrm{abs}}(\varphi) = \sigma\varepsilon(\varphi)T_{\mathrm{sfc}}^4(\varphi) + \operatorname{div}F(\varphi) \qquad (2)$$

in order to disentangle the contributions of changes in absorbed solar radiation, greenhouse warming, and meridional heat transport to the surface temperature evolution of the Hadley states. In this equation, $Q$ is the incoming solar radiation flux at latitude $\varphi$, $\alpha$ is the top-of-atmosphere albedo, $Q_{\mathrm{abs}}$ is the absorbed solar radiation flux (directly diagnosed from model output), $\sigma$ is the Stefan-Boltzmann constant, $\varepsilon$ is the effective emissivity (diagnosed from the ratio of top-of-atmosphere to surface upward long-wave fluxes), $T_{\mathrm{sfc}}$ is the surface temperature, and $\operatorname{div}F$ is the divergence of the total meridional heat transport (diagnosed from the net top-of-atmosphere radiation balance). For a given equilibrium simulation, the surface temperature $T_{\mathrm{sfc,0}}$ can then be derived by simply solving equation (2) for the surface temperature, showing good agreement with the surface temperature directly diagnosed from model output (not shown).

The different contributions to surface temperature changes $\Delta T_{\mathrm{sfc}}$ between a given equilibrium state and a reference state, denoted by subscript 0, can then be calculated as follows:





$$\Delta T_{\text{sfc,solar}}(\varphi) = \left(\frac{1}{\sigma\,\varepsilon_0(\varphi)}\left(Q_{\text{abs}}(\varphi) - \operatorname{div} F_0(\varphi)\right)\right)^{1/4} - T_{\text{sfc,0}}(\varphi) \tag{3}$$

$$\Delta T_{\text{sfc,greenhouse}}(\varphi) = \left(\frac{1}{\sigma\,\varepsilon(\varphi)}\left(Q_{\text{abs,0}}(\varphi) - \operatorname{div} F_0(\varphi)\right)\right)^{1/4} - T_{\text{sfc,0}}(\varphi) \tag{4}$$

$$\Delta T_{\text{sfc,transport}}(\varphi) = \left(\frac{1}{\sigma\,\varepsilon_0(\varphi)}\left(Q_{\text{abs,0}}(\varphi) - \operatorname{div} F(\varphi)\right)\right)^{1/4} - T_{\text{sfc,0}}(\varphi) \tag{5}$$

The results of this exercise are presented in Figure 9 where we show the zonally averaged surface temperature differences between all Hadley critical states and the 900 Ma Hadley state together with the contributions of the three factors.

It is evident from Figure 9 (a) that there is a distinct geographic pattern in the surface temperature changes of the Hadley
states over time: With increasing solar luminosity (i.e. going from 1800 Ma to 0 Ma), there is a gradual cooling in areas covered by sea ice and a (less pronounced) warming in ice-free regions. These trends are driven by the different spatial characteristics of the absorbed solar radiation and the warming due to the greenhouse effect (already noted in earlier work, e.g. Yang et al. 2012a): While the radiative fluxes due to greenhouse gases are distributed relatively uniformly with latitude, the absorbed solar energy shows a marked maximum at the equator and drops off towards the poles, both due to the spatial distribution of the
incoming solar radiation and due to the higher albedos of the ice-covered mid and high latitudes. Going from 1800 Ma to 0 Ma, solar luminosity increases, while the $CO_2$ concentrations of the Hadley critical states decreases, leading to the evolution of their respective contributions to the surface temperature trends shown in Figure 9 (b) and (c). These changes are only partially compensated by adjustments of the meridional heat transport, see Figure 9 (d).

The surface-temperature evolution of the Hadley states described above also provides a qualitative explanation why Hadley
states cannot be stable at low solar luminosities: Going back in time, the decreasing solar radiation is compensated by an increasing greenhouse warming to prevent global glaciation, but the different spatial patterns of these factors lead to progressively shallower surface temperature gradients across the Hadley-cell boundary. At some point, the surface energy budget within the Hadley cell is insufficient to melt sea ice before it is pushed towards the equator by the trade winds, triggering global glaciation due to the ice-albedo feedback. This is also in line with the observed shortening of the transient Hadley phase with decreasing
solar luminosity in simulations on track to a Snowball state shown in Figure 4.

## 4  Discussion and conclusions

In this paper, we have investigated the Snowball bifurcation point (in terms of atmospheric $CO_2$ concentration) of Aquaplanets under the steady increase of solar luminosity over Earth's history in one consistent model framework. We find that until about 1 billion years ago the critical $CO_2$ concentration decreases more ore less logarithmically (from $\sim 10^5$ ppm 4 billion years ago
to $\sim 10^2$ ppm 700 million years ago), as one would expect from the logarithmic character of the $CO_2$ forcing. In more recent





times, critical values decrease more strongly, mainly due to the increasing low-latitude insolation making sea-ice formation more difficult and due to the baseline greenhouse effect of water vapour. We have also put these new simulation results in context by presenting a comprehensive synthesis of proxy $CO_2$ estimates and findings from earlier modelling studies.

The second important new conclusion from our work is a regime shift in critical-state properties about 1.2 billion years ago: While the coldest not fully glaciated climate states at earlier times (and thus lower solar luminosities) are characterised by relatively high global mean temperatures of about $-2°C$ and a sea-ice margin close to the centre of the Ferrel cell ("Ferrel states"), critical states at later times (and thus higher solar luminosities) exhibit much lower global mean temperatures of roughly $-15°C$ and a sea-ice margin at the Hadley-cell boundary ("Hadley states"). These states result from the interplay of the surface energy balance and atmospheric dynamics: In the Ferrel states, the sea-ice margin is stabilised by the poleward push of the meridional winds, whereas in the Hadley states the sea-ice margin has to be maintained by a steep temperature gradient across the Hadley-cell boundary to prevent sea-ice being pushed towards the equator by the trade winds within the Hadley cells. The ultimate cause for the two distinct critical-state regimes are the different spatial distributions of solar radiation and greenhouse-gas forcings: With increasing solar luminosity and decreasing $CO_2$ concentrations, this difference in the spatial distributions will result in ever steeper temperature gradients, making Hadley states stable at some point.

While our investigation of the glaciation threshold through time in one consistent three-dimensional modelling framework and the regime shift from Ferrel to Hadley states are novel, aspects of our work tie in well with earlier findings. The importance of the Hadley circulation for global glaciations, for example, has been noted already by Bendtsen (2002) based on simulations with a simple coupled model. Furthermore, in their studies of modern Snowballs with CCSM3/CCSM4, Yang et al. (2012a, b, c) find two different critical positions for the ice line depending on whether they reduce the solar constant at modern $CO_2$ concentrations or the $CO_2$ concentrations at 94% of the present-day solar constant.

We would like to point out that both the values for the Snowball bifurcation points for the different time slices and the solar luminosity at which the regime shift from Ferrel to Hadley states occurs will depend on model physics (e.g. the snow and cloud schemes) and model parameters (e.g. the cryosphere albedos) and will thus most likely differ between models. Furthermore, we emphasise that aspects of our study are certainly affected by model limitations like the coarse spatial resolution, the simple cloud scheme, or possible deviations of the radiative transfer at very low or very high $CO_2$ concentrations. Most importantly, however, the Hadley cells have a prescribed annual mean width of $30°$ in our simplified atmosphere model, whereas it is well known that they become narrower in colder climate states (e.g. Frierson et al., 2007).

Despite these limitations, we consider the main finding of our paper, the regime shift from Ferrel states at lower solar luminosities to Hadley states at higher solar luminosities, robust. Indeed, there are indications in studies with more complex models supporting this conclusion. Most importantly and as mentioned above, Yang et al. (2012a) and Yang et al. (2012c) investigate modern Snowballs with CCSM3 and CCSM4, respectively, and find that there are no stable states with global sea-ice fractions between $\sim 40\%$ and $\sim 60\%$ for a present-day continental configuration. Note that their values for the critical global sea-ice fractions are somewhat higher than ours (33% and 50%, see Sect. 3.2), which could be partly be due to the differences in boundary conditions, but most likely reflects the fact that in cold climate states the Hadley cells will be narrower than the fixed annual-mean width of $30°$ used in our simplified atmosphere model (see above).



Moreover, Yang et al. (2012a) also report different critical states in CCSM3 depending on the path to global glaciation: For a reduction of the solar constant at present-day $CO_2$ levels, their model enters a Snowball state beyond a global sea-ice fraction of $\sim 40\%$, corresponding to a Ferrel state in our simulations. For a reduction of the atmospheric $CO_2$ concentration at 94% of the present-day solar constant (mimicking Neoproterozoic boundary conditions) on the other hand, the critical sea-ice

fraction is $\sim 60\%$, similar to a Hadley state in our simulations. Yang et al. (2012a) attribute this difference in the paths to global glaciation to the dissimilar spatial characteristics of solar radiation and $CO_2$ forcing, in line with our findings above. Looking at Figure 1, this would put the regime shift from Ferrel to Hadley states in Yang et al. (2012a) to somewhere between $\sim 1.2$ and $\sim 0.7$ billion years ago, in excellent agreement with our findings despite the different boundary conditions. CCSM4 is more sensitive with respect to global glaciation than CCSM3 (Yang et al., 2012c), and in this model the critical state is a Hadley state

for both paths to global glaciation. This would put the transition from Ferrel to Hadley states to times earlier than $\sim 1$ billion years ago, again in principal agreement with our results.

One conclusion from our work is that the Snowball bifurcation (and thus one of the most important limits to planetary habitability) is determined by the interplay of the energy balance and internal dynamics, implying the need for knowledge of certain system parameters and for three-dimensional models in order to be able to assess planetary habitability. Note that

our results cannot be easily generalised to other planets for three reasons: First, the idealised Aquaplanet configuration is a rather special case. Second, our simulations have been performed for one particular orbital configuration. And third, the results will likely depend on the planet's rotation rate since the structure of the Hadley circulation, for example, changes significantly with the rotation rate (e.g. Schneider, 2006). Sampling the parameter space more completely remains to be done in future work. Finally, our results highlight once again the crucial importance of sea-ice dynamics for investigations of the Snowball

bifurcation point, for example in the context of the Faint Young Sun Paradox or the Paleoproterozoic and Neoproterozoic glaciations.

*Code and data availability.*  Model input and output files as well as the scripts used to generate the figures are available at the institutional repository of the Potsdam Institute for Climate Impact Research (reserved DOI 10.5880/PIK.2022.003)[1] Feulner et al. (2022). The model source code is made available upon request.

*Author contributions.*  G.F. designed the study; M.S.B. and G.F. performed and analysed model simulations; S.P. provided technical assistance and compiled the data archive, G.F. prepared all figures; G.F. wrote the paper with input from all co-authors.

*Competing interests.*  The authors declare that they have no conflict of interest.

[1]preview URL: https://www.pik-potsdam.de/data/doi/10.5880/PIK.2022.003



*Acknowledgements.* The authors would like to thank Julius Eberhard, Alexey V. Eliseev and Anna Feulner for help and discussions. The European Regional Development Fund (ERDF), the German Federal Ministry of Education and Research and the Land Brandenburg are
gratefully acknowledged for supporting this project by providing resources on the high performance computer system at the Potsdam Institute for Climate Impact Research.



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

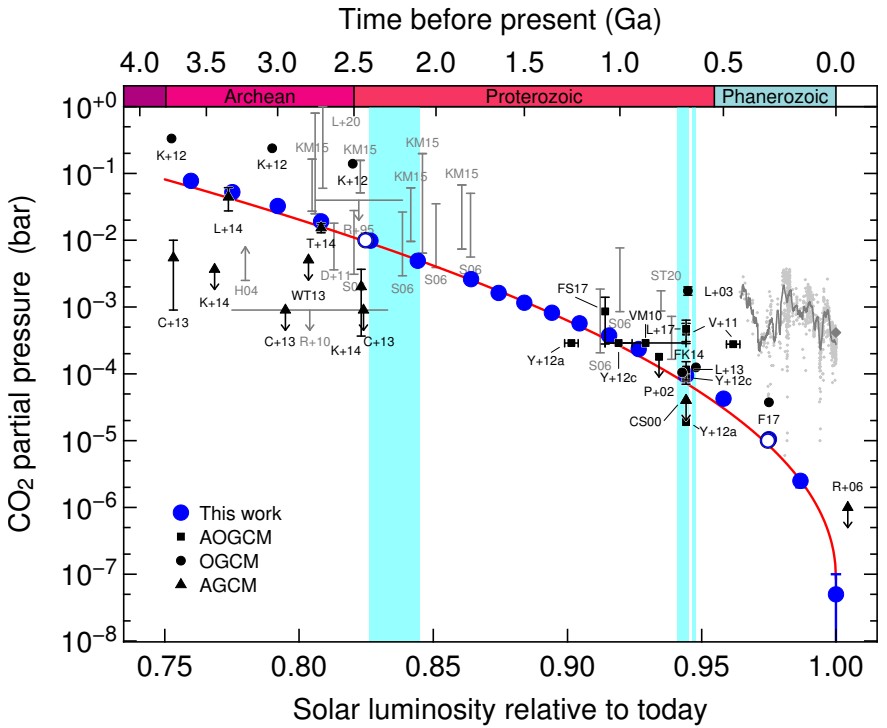

**Figure 1.** Snowball bifurcation (in terms of $CO_2$ partial pressure assuming 1 bar total atmospheric pressure) as a function of solar luminosity for the Aquaplanet simulations presented in this work (large blue circles). The open circles indicate the bifurcation points in solar luminosity for the additional model experiments at fixed levels of 10,000 ppm and 10 ppm of $CO_2$. The red line shows a fit of the function defined in Equation (1) to these bifurcation points, see the main text for details. The smaller black symbols denote earlier modelling studies with AOGCMs (squares), OGCMs coupled to a simplified atmosphere model (circles) and AGCMs coupled to a simplified ocean model (triangles). The model results are averages between the highest $CO_2$ concentration (or solar constant) of fully glaciated states and the lowest value of states with open water, with the range between these two values indicated by the error bars unless the latter are smaller than the symbols. Note that some model studies provide upper limits (indicated by arrows) rather than ranges. In cases where contributions of other greenhouse gases (in particular methane) were included in the models we show the effective $CO_2$ concentration calculated from the combined forcing. All model results are scaled to a modern solar constant of $1361 \, \mathrm{W\,m^{-2}}$ (Kopp and Lean, 2011). Estimates of past atmospheric $CO_2$ levels are shown in grey; Phanerozoic $CO_2$ values are taken from Foster et al. (2017), the modern value is shown as a diamond. Global glaciations in Earth's history (cyan shading) occurred during the Paleoproterozoic and Neoproterozoic eras.

Proxy data: R95 – Rye et al. (1995), H04 – Hessler et al. (2004), S06 – Sheldon (2006), R10 – Rosing et al. (2010), D11 – Driese et al. (2011), KM15 – Kanzaki and Murakami (2015), L+20 – Lehmer et al. (2020), ST20 – Strauss and Tosca (2020).

Model studies: CS00 – Chandler and Sohl (2000), L+03 – Lewis et al. (2003), P+02 – Poulsen et al. (2002), R+06 – Romanova et al. (2006), VM10 – Voigt and Marotzke (2010), V+11 – Voigt et al. (2011), K+12 – Kienert et al. (2012), Y+12a – Yang et al. (2012a), Y+12c – Yang et al. (2012c), C+13 – Charnay et al. (2013), L+13 – Liu et al. (2013), L+14 – Le Hir et al. (2014), WT13 – Wolf and Toon (2013), FK14 – Feulner and Kienert (2014), K+14 – Kunze et al. (2014), T+14 – Teitler et al. (2014), F17 – Feulner (2017), FS17 – Fiorella and Sheldon (2017), L+17 – Liu et al. (2017).



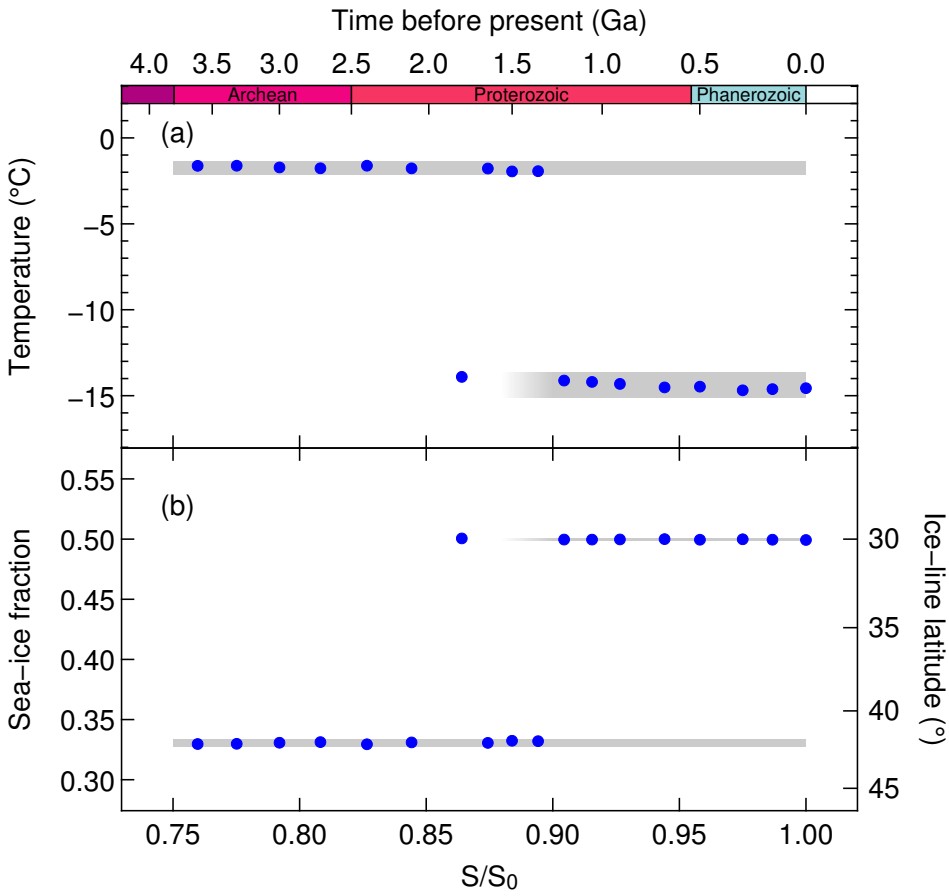

**Figure 2.** (a) Global annual mean surface air temperature and (b) sea-ice fraction of critical states as a function of solar luminosity (blue circles). The right-hand vertical axis in (b) indicates the effective latitude of the critical ice line, calculated from the sea-ice fraction assuming full ice cover over two symmetric polar caps which is a reasonable approximation for the Aquaplanet critical states (see Figure 6). The grey shading indicates the $\pm 3\sigma$ ranges of the respective values. Note that the states with higher temperature and lower sea-ice fraction are also stable states at higher solar luminosities as shown for the 900-Ma time slice in Figure 3.



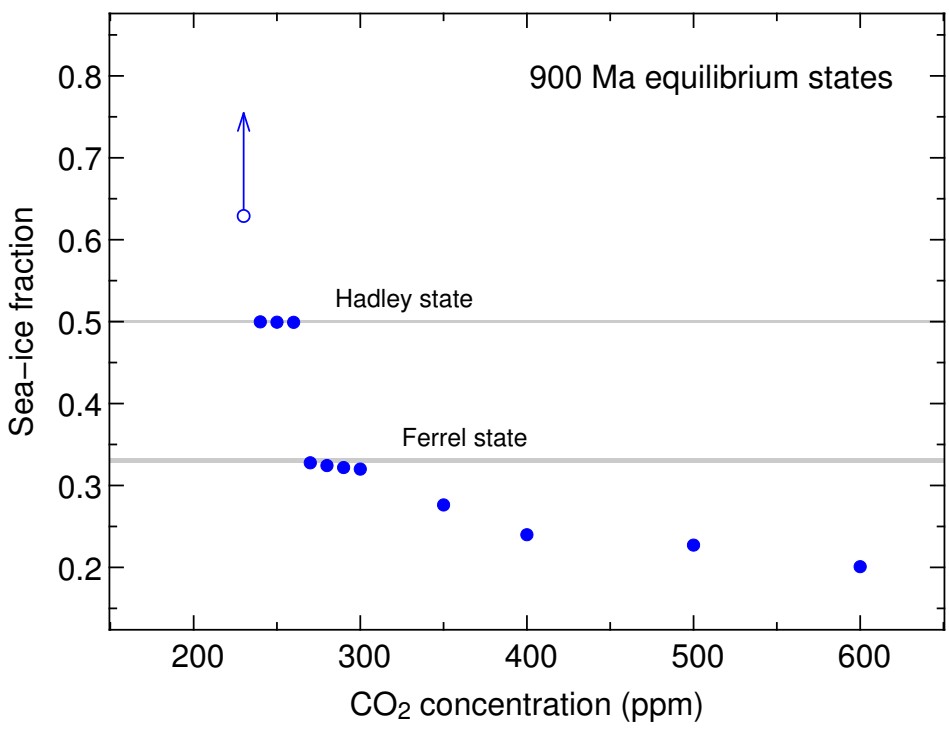

**Figure 3.** Global and annual mean sea-ice fraction in equilibrium climate states at 900 Ma for various concentrations of atmospheric $CO_2$. Grey shaded areas are as in Figure 2. Note that the simulation at 230 ppm is on track to a Snowball state, but has not reached equilibrium due to numerical instability.





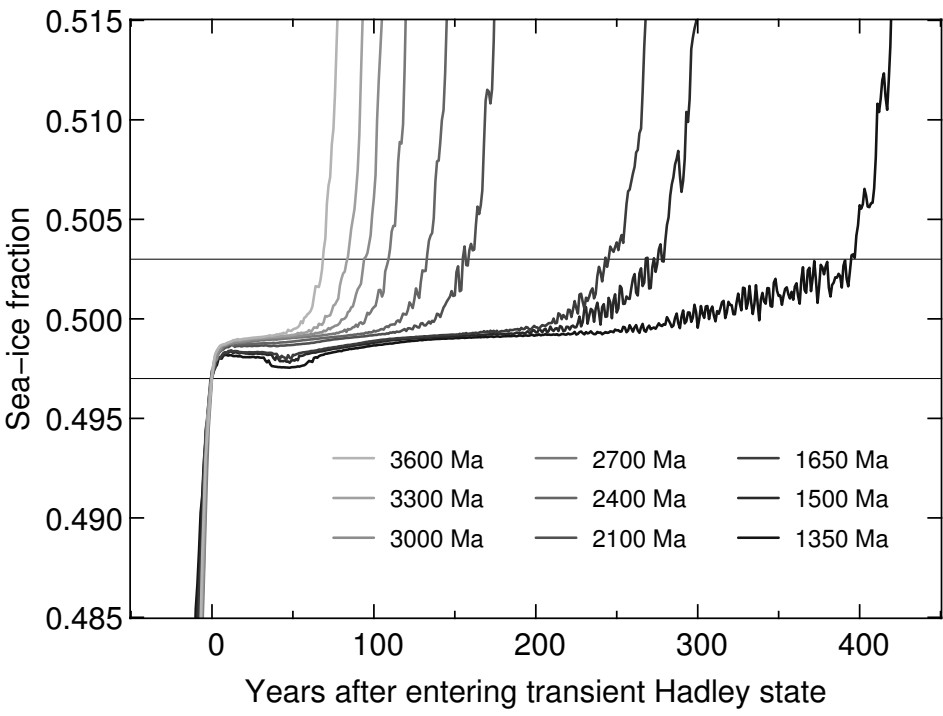

**Figure 4.** Transient Hadley states with a global mean sea-ice cover of about 50% in simulations on track to a Snowball. The transient Hadley phase can be clearly seen as a plateau in all simulations. To capture the entry into and the exit from this state for all time slices, we use a range of $0.497 < f_{\mathrm{sea\,ice}} < 0.503$ to define the transient Hadley phase, indicated by the horizontal lines.



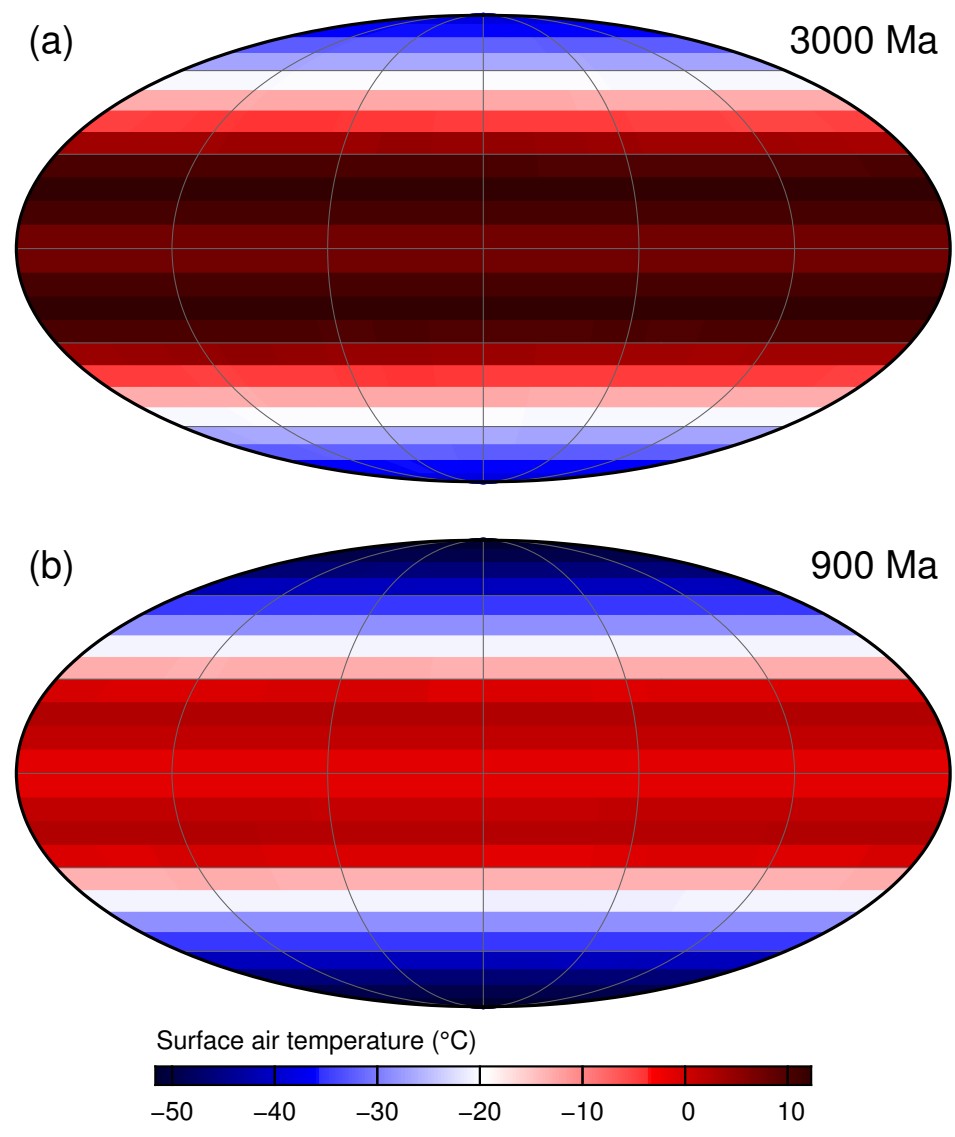

**Figure 5.** Maps of annual mean surface air temperatures for the critical states at 3000 Ma (a) and 900 Ma (b).





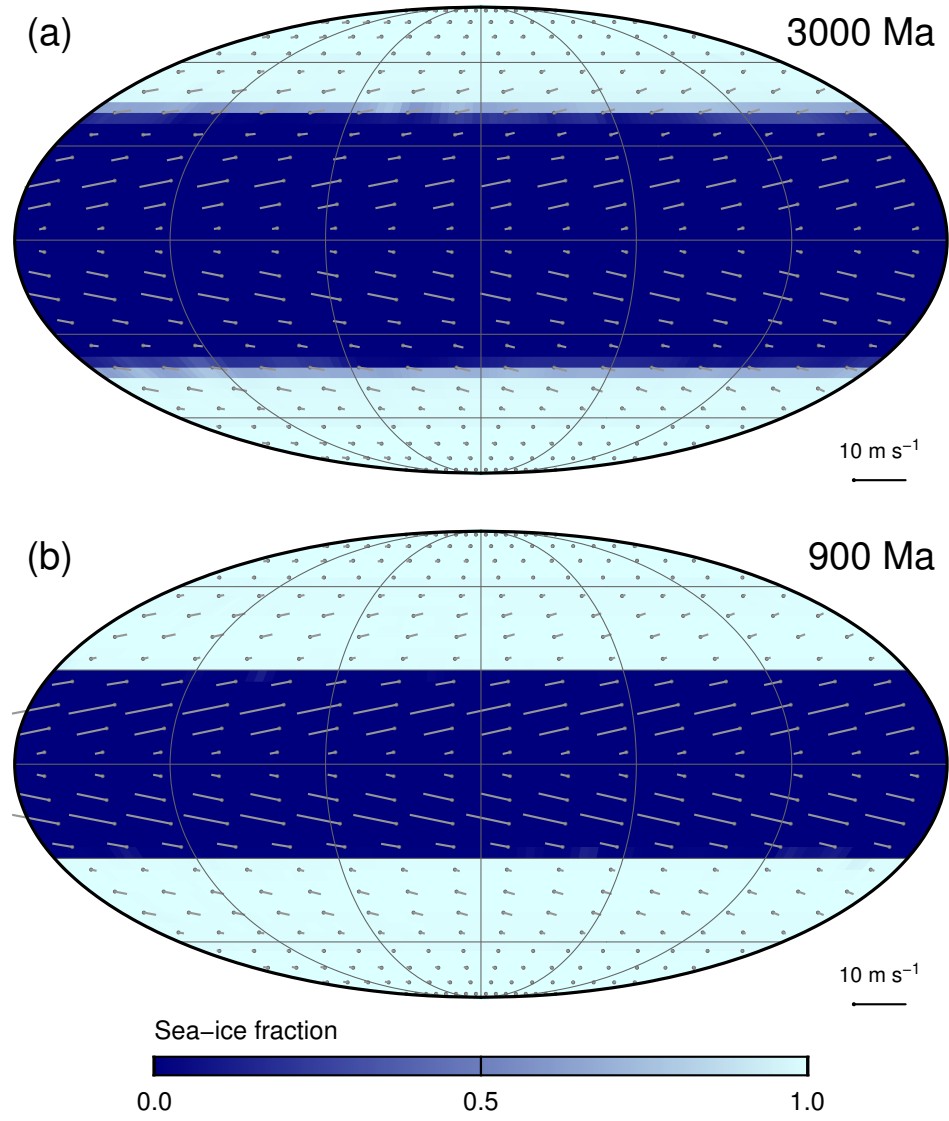

**Figure 6.** Maps of annual mean sea-ice fractions for the critical states at 3000 Ma (a) and 900 Ma (b). Surface wind velocity vectors are shown in the right-hand panels, with the length scaling for a wind speed of 10 m/s indicated in the bottom right corners.



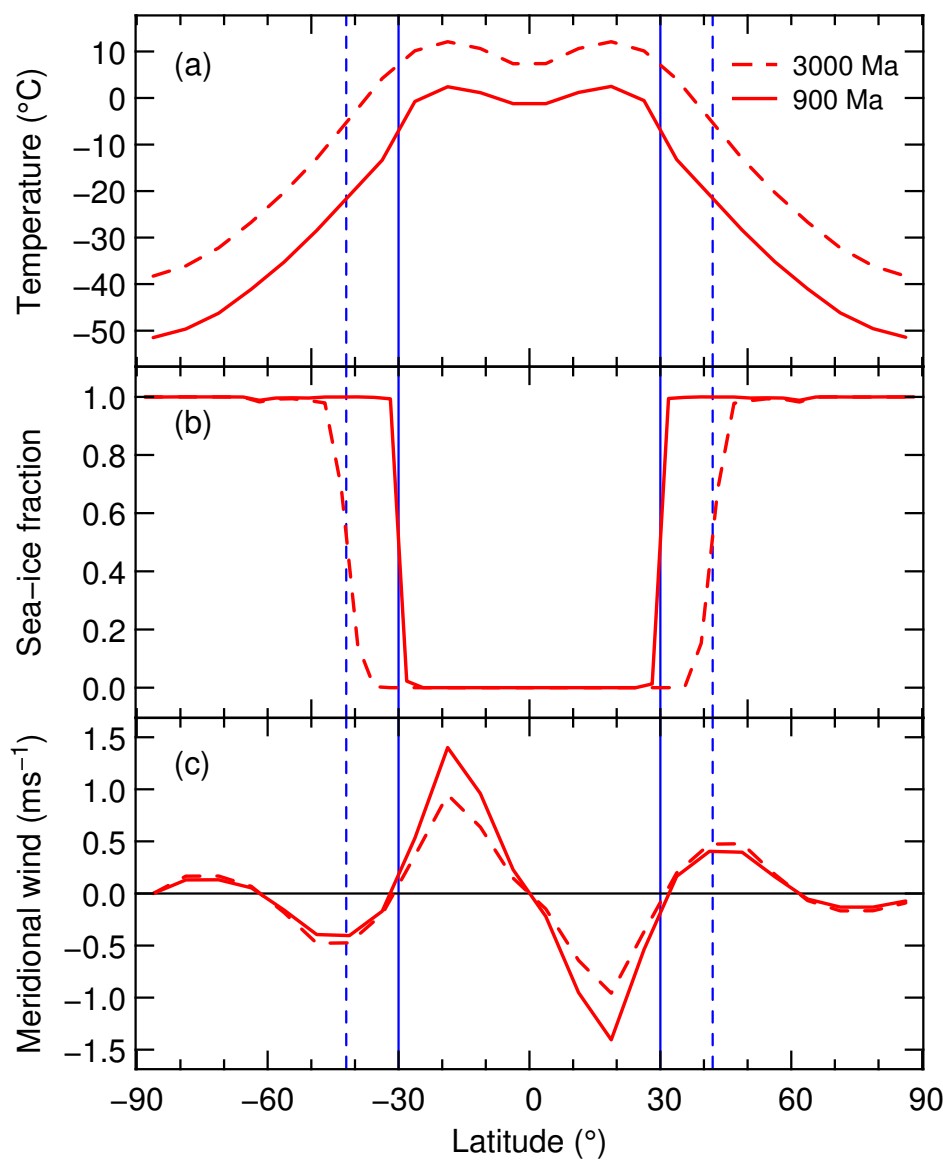

**Figure 7.** Zonal means of (a) annual mean surface air temperatures, (b) sea-ice fractions, and (c) meridional wind speed for the Ferrel state at 3000 Ma (solid red lines) and the Hadley state at 900 Ma (dashed red line). The vertical lines indicate the sea-ice margin for the Ferrel state (dashed blue line) and for the Hadley state (solid blue line).



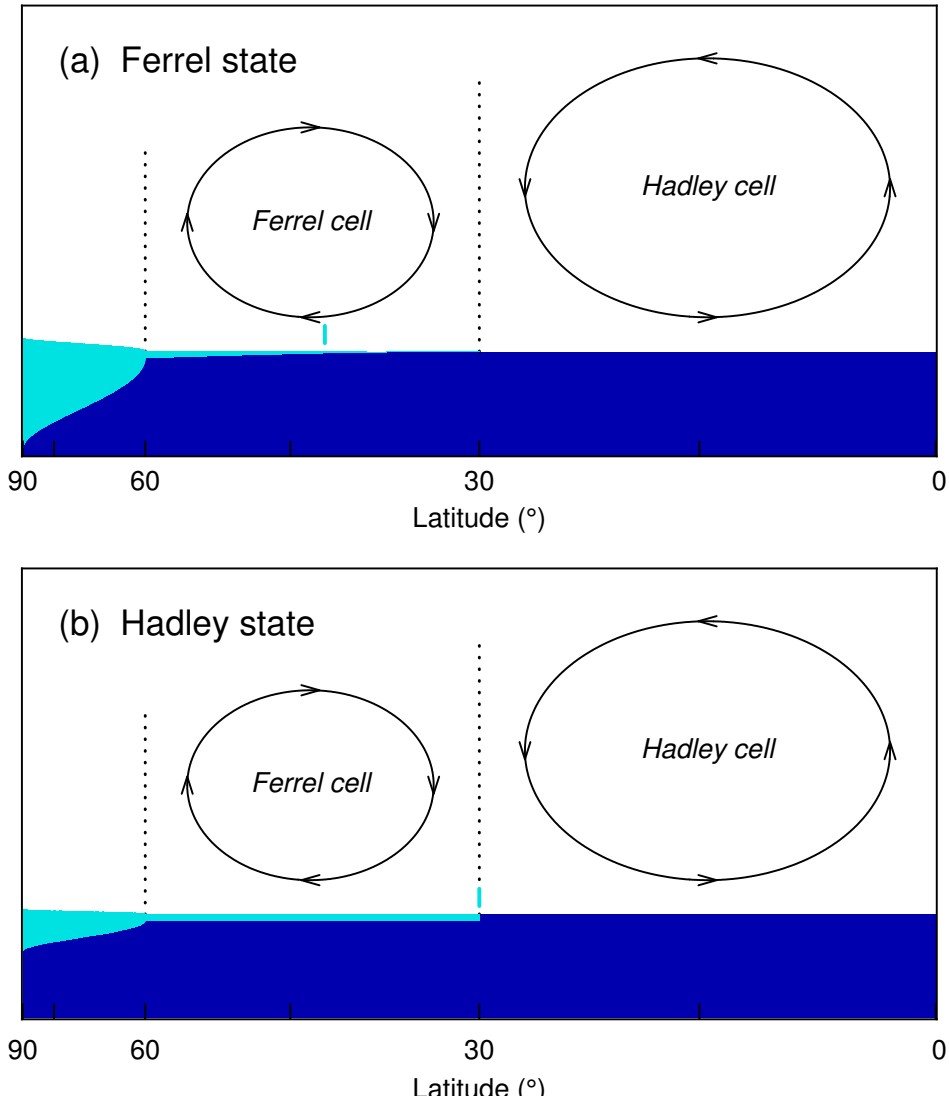

**Figure 8.** Schematic illustration of the differences between the critical states (a) at lower solar luminosities ("Ferrel states") and (b) at higher solar luminosities ("Hadley states") showing sea-ice thickness (cyan, not to scale) on the ocean (blue) and the large-scale atmospheric wind patterns (arrows), see text for discussion. The vertical cyan lines indicate the effective ice-line latitudes calculated assuming symmetric and full ice cover as in Figure 2.

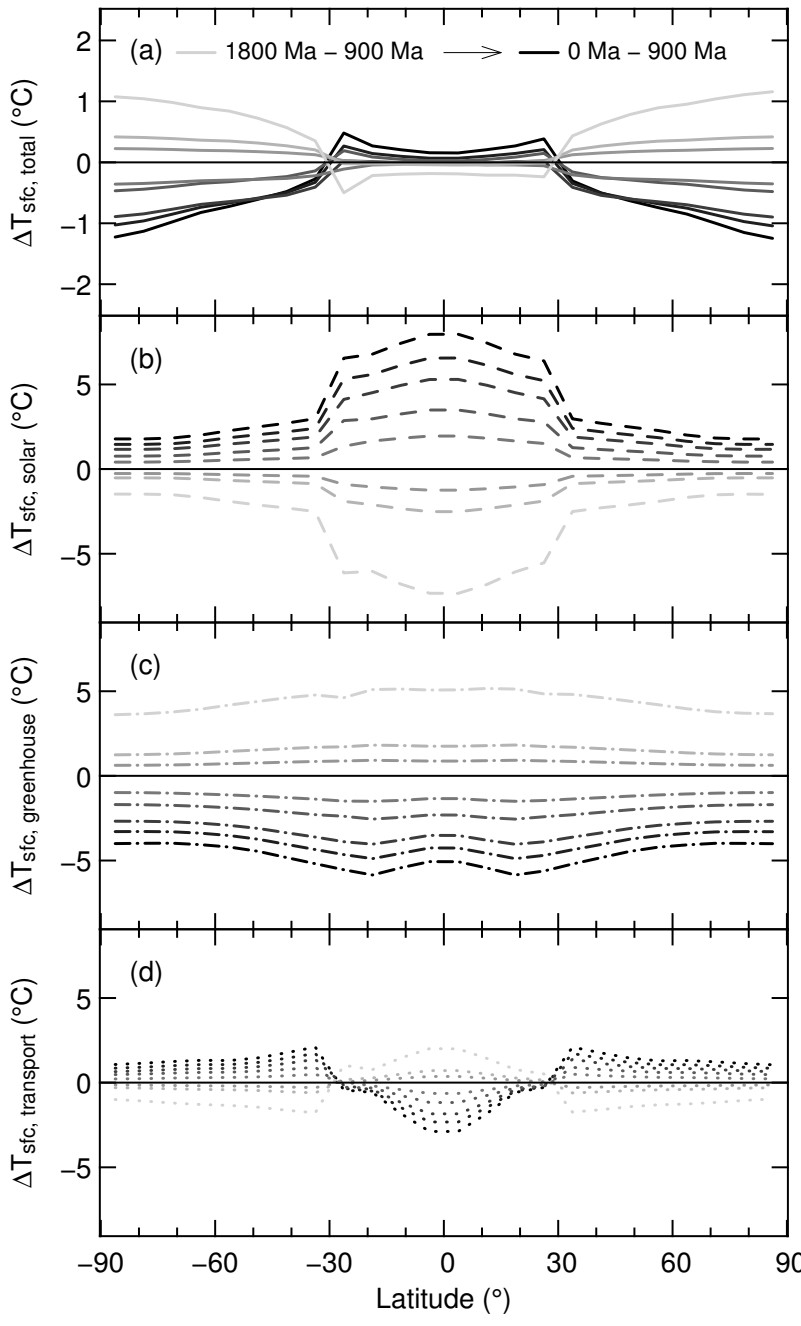

**Figure 9.** (a) Total difference of zonally averaged surface temperatures as diagnosed using a one-dimensional energy balance equation (see text) between the Hadley states from 1800 Ma to 0 Ma and the one at 900 Ma. The other panels show the contributions of changes in absorbed solar radiation (b), greenhouse warming (c), and the combined atmospheric and oceanic meridional heat transport (d). The different Hadley states are indicated by increasingly darker shades of grey for increasing solar luminosities. Note the different scale of the vertical axis in panel (a).