# Peer review of "Tracing the Snowball bifurcation of Aquaplanets through time reveals a fundamental shift in critical-state dynamics"

_Earth System Dynamics, 2022_

## Author Response (AR1)

**Response to reviewers' comments**

This document contains our replies to the comments raised by the two reviewers. Text in grey indicates the reviewers' comments, the black text our response as published in the authors' comments in the public discussion, text in dark blue indicates changes made during revision of the manuscript.

**RC1 by Aiko Voigt**

The authors study the location of the Snowball Earth bifurcation in terms of atmospheric CO2 as a function of insolation in the range of 1361-1034 Wm-2. As the sun becomes stronger over time, the insolation range covers the time from today to 3600 Ma before present, meaning that the work studies the bifurcation as a function of time. The authors apply a model of intermediate complexity with a simplified atmosphere model in aquaplanet setup, which allows them to sweep through a broad range of insolation and CO2 values. Their two main findings are i) that for lower insolation values the critical CO2 decreases logarithmically as insolation increases but drops faster for higher insolation values, and ii) that the nature of critical states (defined as states before the runaway icealbedo feedback sets in) is different between low and high insolation. For low insolation values, the critical ice edge is located in the midlatitudes (termed the Ferrel state by the authors), whereas for higher insolation values it is located in the subtropics (the Hadley state). The authors ascribe this difference in critical states to the meridional gradient in insolation and wind-driven sea-ice transport. The text is well written and the graphics are of high quality (except for two minor questions, see below).

My main criticisms is the following. From reading the text it seems the authors suggest that critical states with a sea ice cover around 50% or with a sea ice edge equatorward of 30 deg were not possible. Yet, there are several studies that have found such states. The conclusion of the change in critical state dynamics thus seems not as robust as described by the authors. I also found that in some cases the comparison with previous studies seems a bit lopsided. I elaborate this below as part of my main comments.

Overall, however, this is a well conducted and well presented paper that addresses a question that was so far not studied. I am confident the authors can address my concerns and recommend minor revisions.

First of all we would like to thank Aiko Voigt for his very constructive as well as helpful comments and the overall positive evaluation of our manuscript. We really appreciate the effort!

Just for clarification, we do not suggest that states with around 50% global sea ice cover are not possible since our Hadley states fall into this category. The existence of

states with a sea ice edge closer to the equator is strongly model dependent, and has also been questioned in a recent paper by Braun et al. (2022) on which the reviewer is one of the co-authors. Our model does not exhibit these states, but we agree that the (still unresolved) question of the existence of these states has to be discussed in the manuscript. By the way, most of the cases where the comparison seems "lopsided" appear to be due to simple misunderstandings, see our detailed response to the individual comments below. As far as these misunderstandings are due to the wording of our manuscript, we are going to improve the text accordingly, again see our replies below.

Main comments:

1. In the conclusion section (L350ff) the authors argue that critical states with a sea ice cover around ∼40% are not possible (the exact numbers are model dependent). The argument is made based on the Ferrel vs. Hadley states, and is allegedly supported by comparison to the work of Yang et al. However, when checking the figures in Yang et al. (2012a) I believe I found some inconsistencies with the authors' arguments. Specifically, Fig. 2 of Yang shows that there are stable states with a sea ice fraction of 50%, contradicting the statement that "... there are no stable states with global sea-ice fractions between ∼ 40% and ∼ 60% for a present-day continental configuration." Probably even more severe, Fig. 16b of Yang et al. (2012a) shows that there is a stable state with 70% sea ice cover for 90% insolation. In my understanding such a state contradicts the Ferrel-Hadley-state argument of the authors. There might be other inconsistencies with the Yang et al results.

Thank you for asking these critical questions. Concerning the first one, we think that it is unclear whether the state with about 50% global sea-ice fraction in Figure 2 of Yang et al. (2012a) will remain stable due to the rather short integration time of this particular model simulation. In fact, the quoted statement is derived from a sentence from Yang et al. (2012a, page 2719, left column) where they write: *In other words, it is likely that there are no stable states between ∼ 40% and 60% sea ice coverage during the initiation of the Snowball Earth; this phenomenon is further confirmed in the simulations with CCSM4 (Yang and Peltier 2012).* Thus there is certainly no inconsistency here.

We have changed the sentence in question in the discussion section, replacing the paraphrased version by a direct quote from Yang et al. (2012a): "Most importantly and as mentioned above, Yang et al. (2012a) and Yang et al. (2012c) investigate modern Snowballs with CCSM3 and CCSM4, respectively, and find that "it is likely that there are no stable states between ∼ 40% and 60% sea ice coverage during the initiation of the Snowball Earth" (Yang et al., 2012a, p. 2719) for a present-day continental configuration."

The second comment is more interesting because Figure 16 in Yang et al. (2012a)

refers to the dependence on initial states of the Yang et al. (2012a) results which we were not aware of. We will discuss and clarify this in the revised version of the paper. We do point out, however, that our statement holds for initialisation from warm climate states corresponding to the procedure used in our model simulations.

We have added the following sentence to Section 2.2: "Note that critical state characteristics might depend on initial conditions (e.g. Yang et al., 2012a); our results are valid for trajectories starting from a warm, ice-free state, other initial conditions are not investigated here."

2. I am missing a discussion about the fact that critical states with sea ice margins quite close to the equator have been found in models, e.g., Voigt and Abbot (2012), Abbot et al. (2011, http://dx.doi.org/10.1029/2011JD015927) and Braun et al. (2022, https://doi.org/10.1038/s41561-022-00950-1). Overall, this makes me think that the changes in the critical state dynamics - although operating in the Climber model used here - are not as robust and fundamental as described by the authors.

As explained above, we are somewhat skeptical regarding the existence of these states. That being said, we agree with the reviewer that the issue should be discussed in the manuscript. We will do so in the revised version. The changes in critical state dynamics should at least be relevant for the majority of models not exhibiting stable waterbelt states; for models with waterbelt states, the Ferrel and Hadley states could be stable states at higher $CO_2$ concentrations (depending on solar luminosity), similar to the situation at higher solar luminosities where both Ferrel and Hadley states can be stable.

We have added a paragraph on critical states with sea-ice margins close to the equator to the introduction of the paper.

Other comments:

L10 and L145: Is the change in the $CO_2$-insolation function related to the change in the critical state dynamics? This is not clear to me.

No, the change in the function is not related to the shift in critical state dynamics. As can be seen from Figure 2, for example, the shift occurs at about 90% of the present-day solar constant, whereas the downturn in Figure 1 is most pronounced beyond 95% of today's solar luminosity.

To clarify this in the revised version, we have amended the abstract adding that the regime shift in the critical states is not related to the downturn of the $CO_2$-insolation function. We have also added the following sentence to the text after the first description of the shift in critical-state dynamics: "Note that the shift in critical-state properties is

not related to the downturn of the bifurcation limit at higher solar luminosities discussed in Sect. 3.1."

L27: It is unclear to me what you mean by "for even lower solar luminosities". What does "even" refer to.

Yes, we understand that the wording could be confusing. We will reword the sentence to make this clear.

The sentence has been reworded: "For solar luminosities and/or greenhouse-gas concentrations lower than today, a bifurcation point in phase space is reached at some point."

L80: Pierrehumbert et al., 2011 (doi:10.1146/annurev-earth-040809-152447) compared Snowball initiation in three AGCMs in aquaplanet setup (their Fig. 4). These models did not include ocean and sea ice dynamics, but used the same coordinated setup. Also, Hoerner et al, JAMES, 2022 (https://doi.org/10.1029/2021MS002734) used an aquaplanet setup to study the impact of sea ice thermodynamics on Snowball initiation. Maybe these are interesting references?

We will include and discuss these additional references in the revised version of the paper.

We have added the following sentence to the introduction where we motivate the use of an Aquaplanet configuration: "Although Aquaplanet setups were used in the context of Snowball glaciations before (e.g. Pierrehumbert et al., 2011; Braun et al., 2022; Hörner et al., 2022), we uniquely focus on the long-term evolution of the bifurcation point."

L101: Some more discussions on the atmosphere model, its limitation and the impacts of its limitations would be desirable. For example, are the Hadley and Ferrel cell boundaries fixed in time, or can they move with the seasonal cycle? How does this impact the P-E patterns and hence snow on sea ice and surface albedo? Do the authors think that this matters? This would also be helpful for the wind argument made around L262 in the result section.

This is an important point. While the annual mean width of the Hadley cells in our simplified atmosphere model is fixed (as we had described it maybe somewhat too briefly in the manuscript), the boundary between the Hadley cells moves with the thermal equator, with a corresponding, but smaller shift in the boundaries between the Hadley and the Ferrel cells, see Petoukhov et al. (2000, Section 3.2). Thus the overall changes of the large-scale circulation with the seasonal cycle are represented in the model in principle. We will add this important information to the description in the revised version of the paper.

The information above has been included in the text of the revised version.

L111: The agreement with the Liu et al (2013) work seems cherry picking and in my view is a weak argument. There are other studies for which the agreement would be much lower, as in fact can be seen from Fig. 1 of the paper.

We agree. What we wanted to convey is that our model gives comparable results for the glaciation threshold to a more sophisticated model with similar cryosphere albedos. However, the sentence in this form was written before the full synthesis presented in Figure 1 was available. We will change this in the revised version of the paper.

The sentence in the paper has been changed as follows: "We note, however, that the Snowball bifurcation points derived for Neoproterozoic time slices with our model (Feulner and Kienert, 2014) fall well within the range of those from state-of-the-art atmosphere-ocean general circulation models (AOGCMs, see also Figure 1) and agree very well with models using similar cryosphere albedo values (Voigt and Abbot, 2012; Yang et al., 2012c; Liu et al., 2013)."

Table 1: I would find it helpful if the S/S0 ratio could be included in the table, as the ratio is used in Figs. 1 and 2.

This is a good idea, we will add an additional column with the ratio to Table 1.

A column with $S/S_0$ has been added to Table 1.

L140 and L193: The 0ppm CO2 value for today's insolation is consistent with Voigt and Marotzke, 2010, who found that removing all CO2 would lead to a Snowball in the coupled ECHAM5/MPI-ESM model (using present-day continents).

Many thanks for the hint, we will add this to the discussion in the revised version of the paper.

We have added the following sentence to the paper:

"Voigt & Marotzke (2010) used ECHAM5/MPI-OM finding a Snowball state at $0.1\,\mathrm{ppm}$ of $CO_2$ for present-day continents and a slightly higher value of the solar constant of $1367\,\mathrm{W\,m^{-2}}$, again in good agreement with our results given the cooling influence of continents and the generally higher susceptibility to global glaciation of their AOGCM."

We have also added the corresponding data point to Figure 1.

L147ff: I do not understand what the authors mean by baseline warming from water

*vapor. I also wonder how clouds are treated in Climber.*

By "baseline warming" we refer to the effect that even in the rather cold, but not fully ice covered states there is evaporation and thus some greenhouse warming due to atmospheric water vapour. We will check whether we can reword the sentence to make this clearer.

The word "baseline" has been deleted to avoid confusion.

The cloud module of CLIMBER-3$\alpha$ uses a two-layer cloud scheme (stratus plus cumulus) with the cloud fractions depending on humidity and vertical velocity, see Petoukhov et al. (2000, Section 3.4).

A brief note on the simple cloud scheme has been added to Section 2.1.

*L162: Voigt et al., 2011, Climate of the Past showed that moving continents to the tropics cools the climate and facilitates Snowball initation. This is in line with the argument made by the authors and maybe worth including.*

We will add this to the discussion in the revised version of our manuscript.

The reference has been added to the text of the revised version.

*L175: I agree with the statement that sea ice dynamics was found to facilitate Snowball initiation. Yet I do not agree that previous studies robustly found that simplified oceans make Snowball initiation more difficult. There are at least three counter examples. Poulsen and Jacob (2004, doi:10.1029/2004PA001056) stated that "The wind-driven ocean circulation transports heat to the sea-ice margin, stabilizing the sea-ice margin.". Rose (2015, https://doi.org/10.1002/2014JD022659) also found a stabilizing role of ocean heat transport. This relates to the argument made in L215 regarding the lack of a full ocean. Voigt and Abbot (2012, https://doi.org/10.5194/cp-8-2079-2012) show explicitly that setting ocean heat transport to zero makes Snowball initiation easier, and they argue that this is related to the subtropical wind-driven ocean cells (see their Figs. 12 and 13).*

You are absolutely correct, of course, and we should and will describe the respective effects of ocean and sea-ice dynamics separately and in more detail in the revised version, see also below.

Several changes have been made in the revised version of the manuscript to make this distinction clearer:
First, results from AGCMs without ocean heat transport have been added to Figure 1, see our response below.

Second, we have added a new paragraph discussing the influence of the ocean heat transport on the glaciation threshold: "**Meridional ocean heat transport makes global glaciation more difficult.** It is evident from Fig. 1 that AGCMs without ocean heat transport consistently show bifurcation points at higher $CO_2$ levels than models with prescribed ocean heat transport or dynamic ocean models. This is in line with earlier findings showing that ocean heat transport towards the sea-ice edge, in particular by the wind-driven ocean circulation, makes Snowball initiation harder (Poulsen and Jacob, 2004; Voigt and Abbot, 2012; Rose, 2015)."

Finally, we have re-worded the paragraph on the importance of sea-ice dynamics by removing all discussion of ocean dynamics: "**Models without sea-ice dynamics are too stable.** Studies carried out with atmospheric general circulation models (AGCMs) coupled to mixed-layer ocean models with prescribed ocean heat transport, but without dynamic sea ice tend to predict lower values for the Snowball bifurcation point (see Fig. 1). Indeed, the fact that models without sea-ice dynamics are artificially stable with respect to the Snowball bifurcation has been noted before (Lewis et al., 2003, 2007; Voigt and Abbot, 2012)."

L180: The study of Pierrehumbert et al., 2011 (see above) tested for albedo values in 3 models, showing that ice albedo differences are key.

We will add this study to the discussion of the impact of cryosphere albedos.

The reference has been added to the revised text.

L198: I believe Lewis et al., 2003 used prescribed surface winds, because of which they could not make robust statements of the impact of sea ice dynamcics. See the discussion of the Lewis work in Voigt and Abbot (2012; page 3 left column).

We are aware that the model used by Lewis et al. (2003) uses prescribed surface winds and will add this to the discussion during revision of the manuscript.

We have amended the sentence on the Lewis et al. (2003) results which now reads: "The reason for the higher value found by Lewis et al. (2003) remains unclear, but could be connected to the fact that surface winds are prescribed in their model."

L261: Is the fuzzy transition a result of seasonal averaging over fully ice covered grid boxes or does the model allow for partially ice covered boxes?

Our sea-ice model allows for partially ice covered grid cells. We will add this information to the model description in the paper.

The following words have been added to the description of the sea-ice model: "... and allowing for partially ice-covered grid cells, ..."

L274: I am wondering about the role of the wind-driven subtropical ocean cells below the Hadley cells. These cells should be represented by the ocean model and are expected to work towards Snowball initiation (see my comment regarding L175).

This is a good point. We fully agree that ocean heat transport makes Snowball initiation more difficult, but preliminary analysis suggests that this effect cannot fully counteract the destabilising effects of sea-ice dynamics. We will expand on this in the revised version of the manuscript.

Following a similar comment in RC2 by Yonggang Liu, we have added comments on the contribution of wind-driven heat transport in the ocean to the temperature gradients across the Hadley-cell boundary in two places: First, we now mention the strengthening of the Ekman transport at higher solar luminosities in the last sentence of the abstract. Second, in the last paragraph of Section 3, we have added the following sentence to the discussion of the decreasing temperature gradients going back in time: "This effect is further enhanced by a weakening of the meridional heat transport with decreasing solar luminosity, driven by a slowdown of the trade winds leading to weaker Ekman transport in the ocean."

Fig. 1: I appreciate the very nice summary of previous modeling work in the figure. Some relevant studies seem to be missing, however. I suggest adding the results of Pierrehumbert et al. (2011), Voigt and Abbot (2012), Hoerner et al. (2022, https://doi.org/10.1029/2021MS002734) and Braun et al. (2022, https://doi.org/10.1038/s41561-022-00950-1). I apologize that these are all studies that I co-authored, I am listing them here since they are missing and I know of them. There might be additional relevant work.

Many thanks for these hints. We had decided against including the Pierrehumbert et al. (2011) results due to the lack of ocean and sea-ice dynamics, and we had not been aware of the papers published in 2022 at the time of submission. The results of these studies and the one by Voigt & Abbot (2012) will be included in the revised version of Figure 1.

As explained briefly in our public response, we had initially decided against adding the values from Pierrehumbert et al. (2011) to Figure 1 worrying about confusion because of the lack of ocean heat transport and the resulting high $CO_2$ levels for the Snowball bifurcation. The same would hold true for both Braun et al. (2022) and Hörner et al. (2022). However, we have now introduced a new symbol for AGCMs without ocean heat transport in Figure 1, so the results from these papers have now been added to the revised version of the Figure. We have also included the results from Voigt & Abbot (2012) as well as the lower limit for the modern Snowball from Voigt & Marotzke (2010).

Are Figs. 5 and 6 needed given the zonal symmetry and the zonal-mean plots in Fig. 7?

Yes, Figures 5 and 6 are somewhat redundant with Figure 7. Our intention was to illustrate the general climate states with maps, which are less abstract than the more aggregated Figure 7. And besides the meridional temperature distribution, these maps also show that the model indeed exhibits the zonal uniformity that is to be expected for an aquaplanet. We will explore different ways to combine the maps with Figure 7 or use Figures 5 and 6 to display seasonal variations.

In order to be able to show the spatial patterns while at the same time avoiding redundancy and saving space, we have now combined the four world maps from the old Figures 5 and 6 into a new Figure 5 where we show only one sector (i.e. one quarter of the world) with the surface air temperatures and sea-ice fractions (plus wind fields) for each of the two typical critical states.

Fig. 8: I do not understand the meaning of the legend in panel a and the color coding of the lines.

The one-dimensional energy balance equation can be used to attribute changes in surface temperature between different equilibrium states, in other words surface temperature differences of two climate states. In Figure 8, we compare the surface temperature differences between the Hadley states for the different time slices and the one at 900 Ma. Thus "1800 Ma − 900 Ma" in the legend is to be read as "1800 Ma *minus* 900 Ma". We tried to explain this in the caption of the Figure, but will attempt to make this clearer during revision.

Presumably, we are talking about Fig. 9 (in the original numbering) rather than Fig. 8, also in our published response, our apologies. We have replaced the legend in panel (a) to now read "1800 Ma vs. 900 Ma" and "0 Ma vs. 900 Ma" to make the meaning clearer.

**RC2 by Yonggang Liu**

Feulner et al. traced the snowball bifurcation of an aquaplanets through the history of the Earth using a climate model of intermediate complexity. To my knowledge, this has never been done before. Importantly, they did get some very interesting results from such practice and found that the critical climate state was different under low and high solar constants. In particular, they found that once the sea-ice edge crosses 40° latitude, it would march forward to the equator without any external forcing when the solar constant is low. While when the solar constant is high, this critical latitude is at 30° latitude, i.e. the boundary of the Hadley cell. Therefore, I think the work provides new knowledge about the stability of the Earth's climate to the society and definitely worth publication. However, there are still a few relatively small things that need to be clarified. Especially, the mechanism for the stability of the 'Hadley state' may be explained better.

First of all we would like to thank Yonggang Liu for his positive evaluation and constructive review of our manuscript for which we are really grateful. Our response to the individual comments can be found below.

The major reason that the ice edge cannot be stabilized at ∼30° latitude when the solar constant is low, I think, is because the atmosphere+ocean heat transport across the 30° latitude exceeds the energy that can be received by the oceans equatorward. This will cause a continuous cooling of the tropical region and eventually allows the ice edge to march forward towards the equator. The rate of this cooling should be inversely proportional to the solar constant and is indeed well indicated by their Fig. 4. Therefore, I hope they can demonstrate this mechanism more clearly by showing explicitly the total meridional heat transport at 30°S and 30°N and the solar energy received by the oceans within 30° latitudes. These should be shown for the transient stage in one of the simulations, for example, at year 100 of the 1500 Ma simulation in Fig. 4.

If the point above can be confirmed, then the mechanism for stabilizing the 'Hadley state' may need to be modified (such as the last sentence of the abstract). It is stabilized by enhanced oceanic heat transport once the sea-ice edges approach the boundary of Hadley cell. The enhanced heat transport is expected to be due to stronger easterly winds as normally seen in other models and thus stronger poleward Ekman transport. This mechanism always works as clearly shown in Fig. 4 but it can stabilize the climate only momentarily when solar constant is low because the enhanced heat transport extracts more energy than the tropical ocean can receive; the heat content of the tropical ocean is drained out quickly. While for a high solar constant, a balance can be achieved easily (other feedback processes are naturally involved, especially the outgoing longwave radiation, latent heat flux etc. so that the tropical ocean will lose less energy in these ways) unless the $CO_2$ concentration is lowered further. This is likely also the major reason that the slope in Fig. 1 increases once such balance can

be achieved.

Many thanks for this interesting idea. We will analyse the model diagnostics to check the balance between the surface energy budget within the Hadley cells and the total meridional heat transport at 30°N/S. We will also look into the role of the oceanic heat transport in stabilising the Hadley states and will modify the manuscript accordingly.

As suggested, we have analysed the energy budget and meridional heat fluxes for the transient Hadley state at 1500 Ma and for year 100 after entering the Hadley state. The total solar radiation absorbed by the surface between latitudes 30°N/S is 48.5 PW; however, the more appropriate quantity might be the surface radiation balance (absorbed solar radiation minus outgoing long-wave radiation) which amounts to 30.1 PW over the same region. The total atmospheric meridional heat transport across the Hadley-cell boundaries at 30°N/S is 12.3 PW, its equivalent in the ocean 3.5 PW, resulting in a total meridional heat transport of 15.8 PW out of the tropics. Therefore, we cannot confirm the idea that more energy is transported away than received by the tropical oceans.

We can confirm, however, that the wind speeds within the Hadley cell increase with increasing solar luminosity, leading to enhanced Ekman transport in the oceans towards the sea-ice edge. This is also visible from the energy balance model analysis shown in Figure 8 of the paper, where the warming just within the Hadley cells with increasing solar luminosity is driven by a combination of more absorption of short-wave radiation and stronger meridional heat transport in the ocean. So we still think that the surface energy budget close to the sea-ice edge is the decisive factor in stabilising the Hadley states, but we agree that ocean heat transport plays an important role here (see also the corresponding comment in RC1 by Aiko Voigt).

Therefore, we have modified the manuscript in two places: First, we now mention the strengthening of the Ekman transport at higher solar luminosities in the last sentence of the abstract as suggested. Second, in the last paragraph of Section 3, we have added the following sentence to the discussion of the decreasing temperature gradients going back in time: "This effect is further enhanced by a weakening of the meridional heat transport with decreasing solar luminosity, driven by a slowdown of the trade winds leading to weaker Ekman transport in the ocean."

Minor questions:

How are the poles treated in the ocean module of this model since an aquaplanet is simulated?

In contrast to other coupled model simulations of aquaplanets, we do not place small islands at the poles, so the poles are treated similar to the North pole in present-day simulations with ocean models using spherical grids, applying filtering in the polar

regions in order to be able to increase the ocean model time step without causing numerical instabilities. We will add this information to the revised version of the manuscript.

A sentence on the lack of polar islands and the use of polar filtering has been added to Section 2.1.

Is the boundary of the Ferrel cell also fixed in this model?

In the annual mean, the poleward boundary of the Ferrel cell is indeed at $60°$N/S. However, as for the Hadley cell, there is a small shift in the boundary reflecting the seasonal shifts of the thermal equator and the changes of atmospheric circulation patterns, see Petoukhov et al. (2000, Section 3.2). As promised also in our response to Reviewer 1, we will describe this in more detail in the revised version of the manuscript.

This is now described in more detailed in the revised version, see also our response to RC1 by Aiko Voigt.

Can the oceanic and atmospheric heat transport be calculated separately in this model?

Yes, the heat transport in the atmosphere and the ocean can be diagnosed separately.

No changes to the manuscript are required in our opinion.

L319: ore → or

Thanks for spotting this typo which will be corrected in the revised version of the manuscript.

Corrected.